# Role of Mitochondrial Glycerol-3-Phosphate Dehydrogenase in Metabolic Adaptations of Prostate Cancer

**DOI:** 10.3390/cells9081764

**Published:** 2020-07-23

**Authors:** Alena Pecinová, Lukáš Alán, Andrea Brázdová, Marek Vrbacký, Petr Pecina, Zdeněk Drahota, Josef Houštěk, Tomáš Mráček

**Affiliations:** Department of Bioenergetics, Institute of Physiology, Czech Academy of Sciences, Videnska 1083, 142 20 Prague, Czech Republic; lukas.alan@fgu.cas.cz (L.A.); andreabrazdova@centrum.cz (A.B.); marek.vrbacky@fgu.cas.cz (M.V.); petr.pecina@fgu.cas.cz (P.P.); zdenek.drahota@fgu.cas.cz (Z.D.); houstek@biomed.cas.cz (J.H.)

**Keywords:** GPD2 gene, mitochondrial glycerol-3-phosphate dehydrogenase (EC:1.1.5.3), prostate cancer, metabolic adaptation

## Abstract

Prostate cancer is one of the most prominent cancers diagnosed in males. Contrasting with other cancer types, glucose utilization is not increased in prostate carcinoma cells as they employ different metabolic adaptations involving mitochondria as a source of energy and intermediates required for rapid cell growth. In this regard, prostate cancer cells were associated with higher activity of mitochondrial glycerol-3-phosphate dehydrogenase (mGPDH), the key rate limiting component of the glycerophosphate shuttle, which connects mitochondrial and cytosolic processes and plays significant role in cellular bioenergetics. Our research focused on the role of mGPDH biogenesis and regulation in prostate cancer compared to healthy cells. We show that the 42 amino acid presequence is cleaved from N-terminus during mGPDH biogenesis. Only the processed form is part of the mGPDH dimer that is the prominent functional enzyme entity. We demonstrate that mGPDH overexpression enhances the wound healing ability in prostate cancer cells. As mGPDH is at the crossroad of glycolysis, lipogenesis and oxidative metabolism, regulation of its activity by intramitochondrial processing might represent rapid means of cellular metabolic adaptations.

## 1. Introduction

Prostate cancer is the second most common cancer in men and the sixth leading cause of death worldwide [1]. Unlike a variety of solid tumors that increase glucose uptake to cover higher demands for biosynthetic compounds [2], prostate cancer cells have a different phenotype [3]. In normal prostate epithelial cells, the oxidation of citrate is impaired mainly due to the inhibition of mitochondrial aconitase (by high levels of zinc) and the accumulated citrate is then released into the lumen [4]. Upon malignant transformation, levels of citrate and zinc decrease [4] due to reactivation of aconitase and subsequent citrate oxidation in the Krebs cycle [5] or citrate-derived de novo lipid synthesis [6]. In both processes, a crucial role is played by mitochondria, organelles responsible for metabolic plasticity of many cancer types [7,8,9].

As the prognosis for advanced and relapsing patients is poor [10], researchers recently focused on targeting metabolic rearrangements in prostate cancer [11,12,13,14,15,16]. The metabolic adaptations were associated with different pathways including oxidative phosphorylation [13,16], lipid metabolism [15] or glucose utilization [14], i.e., quite diverse portfolio of potential targets. Investigation of metabolic characteristics of prostate benign and malignant cell lines [13] showed that cancer cells have reduced glycolytic activity and more active mitochondria (high glucose-driven respiration in intact cells). In contrast to prostate cancer, benign cells preferred fatty acids as an energy source.

Prostate cancer cells were also associated with higher activity of mitochondrial glycerol-3-phosphate dehydrogenase (mGPDH) located on the outer face of the inner mitochondrial membrane [17,18]. mGPDH catalyzes the rate limiting step of glycerophosphate shuttle involved in oxidation of cytosolic NADH. It converts glycerol-3-phosphate to dihydroxyacetone phosphate and concurrently the two electrons are transferred to coenzyme Q pool (reviewed in [19]), and was shown to act as a potent producer of reactive oxygen species in mitochondria [20,21]. mGPDH is also involved in the regulation of cytosolic glycerol-3-phosphate availability for triglyceride and phospholipid synthesis [22]. The mGPDH levels vary across different tissues and its expression is high in many tumors [17,23,24].

In this paper, we compare energy metabolism of prostate benign and malignant cell lines and demonstrate that the important role in oxidative metabolism of cancer cells is played by mGPDH. An observed increase in enzyme activity is not only connected to increased protein levels, but also to more pronounced processing of the enzyme from a precursor into the mature form. Furthermore, increased mGPDH content leads to increased invasiveness, demonstrated as faster wound-healing. We suggest that mGPDH may be important for metabolic adaptations in prostate cancer cells.

## 2. Materials and Methods

### 2.1. Cell Culture

All cell lines were cultivated under standard conditions (37 °C, 5% CO_2_ atmosphere). The normal human immortalized prostate epithelial cell line PNT1A (95012614, Merck/Sigma, Darmstadt, Germany) and androgen-sensitive human prostate cancer cell line LNCaP (kindly provided by Dr. Hodny, IMG CAS, Prague, Czech Republic) and human embryonic kidney cells (HEK293) were purchased from ATCC (CRL-1573, ATCC, Manassas, VA, USA). Prostate cell lines were cultivated in RPMI 1640 medium (Thermo Fisher, Waltham, MA, USA) and HEK293 in high glucose DMEM (Thermo Fisher), both media were supplemented with 10% (*v*/*v*) fetal bovine serum (Thermo Fisher) and antibiotics (100 U/mL penicillin and 100 μg/mL streptomycin, Thermo Fisher).

### 2.2. Models of mGPDH Overexpression

GPD2 (CCDS2202.1) cDNA was obtained by reverse transcription of RNA isolated from HEK293 cells, followed by PCR using Q5 High-Fidelity DNA polymerase (New England Biolabs, Ipswich, MA, USA) and primers: GPD2fw 5′ GGAGGTACAGAGTAGGAGAAG, GPD2rev 5′ CTCACAATCCTCCACAACTAC. By subsequent PCR we added the overhangs containing restriction enzymes NheI and EcoRI (New England Biolabs) to clone GPD2 without any tag into the pcDNA3.1^(+)^ (Thermo Fisher; GPD2pcDNA3fw ACCCAAGCTGGCTAGCGCCA-CCATGGCATTTCAAAAGGCAGTGAAAG, GPD2pcDNA3rev GATATCTGCAGAATTCCTCAC-AATCCTCCACAACTACGGTC) creating a vector with an untagged form of GPD2 (oe). The C-terminal FLAG tag was added to the GPD2 sequence by the reverse primer (GPD2pcDNA3revFLAG GGGCCCTCTAGACCTTGTCGTCATCGTCTTTGTAGTCAAGAATTCCCAATC) and then using the XbaI restriction enzyme inserted into pcDNA3.1^(+)^. The new open reading frame was called the C-term. Then we introduced the SacI restriction site in the position of Ala200 and used the two SacI restriction sites present (the first original one on plasmid and the second 600 bp downstream of the initiation codon—inside the GPD2 gene) for targeted insertion. Using GeneArt Strings DNA Fragment synthesis (Thermo Fisher) we designed three GPD2 variants with a FLAG tag at different positions. We obtained GPD2 with a FLAG tag following initial start codon (N-term), GPD2 with a FLAG tag following Phe27 (27AA) and GPD2 with a FLAG following Ala42 (42AA). All of the vectors were sequenced to confirm proper cloning. Control vectors or vectors containing GPD2 with or without FLAG were transfected into HEK293. LNCaP cells were transfected with an empty pcDNA3.1 vector or with a pcDNA3.1 vector containing untagged GPD2. HEK293 and LNCaP cells were transfected using Metafectene Pro (Biontex, Munich, Germany). Optimal transfection ratio of plasmid DNA to Metafectene was 1:4 (*w*/*v*) (HEK293) or 1:3 (*w*/*v*) (LNCaP). Cells were transfected either transiently (HEK293) or stable transfectants were selected using 1 mg/mL (HEK293) or 0.8 mg/mL (LNCaP) G418 (Merck/Sigma).

### 2.3. Electrophoretic Analyses

Samples for SDS polyacrylamide gels (PAGE) were denatured for 10 min at 60 °C in a sample lysis buffer containing 50 mM Tris-HCl pH 7.0, 4% (*w*/*v*) SDS, 10% (*v*/*v*) glycerol and 0.1 M 1,4-dithiothreitol and separated on a 12% polyacrylamide minigels (MiniProtean III, Bio-Rad, Hercules, CA, USA) using the Tricine buffer system [25]. To achieve better separation, for analysis of HEK293 overexpressing mGPDH-FLAG, midi gel system Hoefer SE 600 (Thermo Fisher) was used.

For native electrophoresis, cells were centrifuged 10 min at 600× *g* and resulting pellets were resuspended in buffer containing 50 mM NaCl, 2 mM 6-aminohexanoic acid, 50 mM imidazole and 1 mM EDTA, pH 7.0. Proteins were solubilized with digitonin (2 g/g protein) for 10 min on ice and centrifuged for 20 min at 30,000× *g* to remove cell debris. Ponceau red dye (0.01%) and 10% glycerol were added to the supernatants and the samples were analyzed by high-resolution clear-native electrophoresis 3 (hrCNE, [26]) using 5–13% polyacrylamide gradient mini gels (MiniProtean III, Bio-Rad).

For two-dimensional (2D) analyses, strips of the first dimension gel (hrCNE) were incubated in 2D buffer containing 1% (*w*/*v*) SDS and 1% (*v*/*v*) 2-mercaptoethanol for 1 h at room temperature and then resolved in the second dimension on SDS-PAGE as described above.

### 2.4. In-gel Activity Staining of mGPDH

Activity staining of mGPDH in native gels was performed according to a modified protocol originally described in [27]. Gel slices were stained using solution of 5 mM Tris–HCl (pH 7.4), 3 mM MgCl_2_, 0.88 mM menadione, 1.2 mM NitroBlue Tetrazolium, 1.5 μM rotenone, 2 mM KCN and 10 mM glycerol-3-phosphate (Merck/Sigma, 61668) for 1 h. Subsequently gels were denatured in 50% methanol/10% acetic acid for 15 min, fixed in 10% acetic acid for 10 min and scanned on a flatbed scanner.

### 2.5. Western Blot Analysis

Gels were blotted on to a PVDF (polyvinylidene difluoride) membrane (Immobilon FL 0.45 μm, Merck) by semi-dry electro-transfer (1 h at 0.8 mA/cm^2^) using a Transblot SD apparatus (Bio-Rad). The PVDF membrane was washed for 5 min in TBS (150 mM Tris- HCl and 10 mM NaCl; pH 7.5) and blocked in 5% (*w*/*v*) fat-free dry milk diluted in TBS for 1 h. Then, the membrane was washed 2 × 10 min in TBST (TBS with 0.1% (*v*/*v*) detergent Tween-20). For immunodetection, the membrane was incubated in a primary antibody (2 h at room temperature or overnight at 4 °C). They included antibodies to actin (MAB1501, Merck/Sigma), FLAG (F1804, Merck/Sigma), citrate synthase (129095, Abcam, Cambridge, United Kingdom), complex I (NDUFA9: 14713, Abcam), complex II (SDHA: 14715, Abcam), complex III (UQCRC2: 14745, Abcam), complex IV (COX4: 14744, Abcam), complex V (ATP5A: 14748, Abcam) and IMMP2L (ARP60563_P050, Aviva Systems Biology, San Diego, CA, USA). The rabbit polyclonal antibody to porin was a kind gift from Professor Vito de Pinto (Dipartimento di Scienze Chimiche, Catania, Italy), and the rabbit polyclonal antibody to mGPDH was custom prepared [28]. For quantitative detection, the corresponding infrared fluorescent secondary antibodies (Alexa Fluor 680, Thermo Fisher; IRDye 800, LI- COR Biosciences, Lincoln, NE, USA) were used. Detection was performed using the fluorescence scanner Odyssey (LI-COR Biosciences) and signals were quantified by ImageLab 6.0 software (Bio-Rad).

### 2.6. Proteomic Analysis

mGPDH fused with C-terminal FLAG was affinity enriched from digitonin (2 g/g protein) HEK293 cellular lysate using anti-FLAG M2 magnetic beads (M8823, Merck/Sigma). Washed beads with immunocaptured proteins were digested “on beads” with trypsin using a sodium deoxycholate procedure as described in [29]. Desalted peptide digests were analyzed in Orbitrap Fusion (Thermo Fisher) mass spectrometer. Resulting raw files were processed by MaxQuant with enabled “dependent peptides” feature [30] at a false discovery rate of 0.01.

### 2.7. Enzyme Activity Assay

Activity of mitochondrial glycerol-3-phosphate dehydrogenase was determined spectrophotometrically as glycerol-3-phophate: 2,6-dichlorophenolindophenol oxidoreductase (GP:DCPIP, ε610 = 20.1 /mM/cm) monitored at 610 nm. The assay medium contained 50 mM KCl, 10 mM Tris–HCl, 1 mM EDTA, 1 mg/mL BSA and 1 mM KCN, pH 7.4. The reaction was started by adding 25 mM glycerol-3-phosphate and 100 μM DCPIP. Absorbance changes were monitored for 60 s at 30 °C. Enzyme activity was expressed as nmol/min/mg protein.

### 2.8. Oxygen Consumption and Extracellular Acidification

The Seahorse XFe24 Analyzer (Agilent, Santa Clara, CA, USA) was used to measure respiratory and glycolytic activity in intact prostate cell lines as described in [31]. The 3 × 10^4^ cells were seeded 1 day prior to the measurement in the culture medium. The measurement was performed in 0.5 mL of the XF medium (modified DMEM: D5030, Merck/Sigma) and oxygen consumption rate (OCR) and extracellular acidification rate (ECAR) were detected after addition of 10 mM glucose, 1 μM oligomycin and 0.5 μM FCCP. Non-mitochondrial oxygen consumption (after inhibition of respiration by 1 μM rotenone and 1 μg/mL antimycin) was subtracted from OCR values and non-glycolytic values (after inhibition of glycolysis by 100 mM 2-deoxyglucose) from ECAR values. Both oxygen consumption rate (OCR) and extracellular acidification rate (ECAR) were normalized to the number of cells stained with 5 μg/mL Hoechst (Thermo Fisher) using Cytation™ 3 cell imaging multi-mode reader (BioTek, Winooski, VT, USA).

### 2.9. Reactive Oxygen Species Production

Reactive oxygen species (ROS) production was estimated as 1 µM CM-H_2_DCFDA (5-(and-6)-chloromethyl-20,70-dichlorodihydrofluorescein diacetate, Thermo Fisher) fluorescence as described in [31,32]. Briefly, cells (2 × 10^5^) were grown in 24-well culture plates in the presence or absence of uncoupler and the formation of the fluorescent compound, dichlorofluorescin, was monitored for 2 h with excitation set to 485 ± 7.5 nm and emission to 535 ± 15 nm using an Infinite M200 plate reader (Tecan Group Ltd., Männedorf, Switzerland). FCCP sensitive ROS production represents part of ROS dependent on the mitochondrial membrane potential and therefore of mitochondrial origin [32].

### 2.10. Cellular Fractionation

Mitochondria from LNCaP cells were released by hypotonic shock and isolated using differential centrifugation as described [33]. In brief, 10% of the cell homogenate was centrifuged at 400× *g* (4 °C, 10 min) and the sedimented nuclear fraction (nuclei) was collected. The supernatant was centrifuged at 10,000× *g* (4 °C, 10 min), the resulting post-mitochondrial supernatant (PMS) was collected and sedimented mitochondria were washed. Aliquots of all fractions were stored at −80 °C.

### 2.11. Scratch Assay

The in vitro scratch assay was performed on LNCaP cells stably transfected with the control pcDNA3.1^(+)^ vector or vector containing GPD2 (untagged form) according to [34]. Cells were seeded at the density of 3 × 10^6^ cells/well in 6-well plate and maintained in the cell culture medium. The plates were kept in 5% CO_2_ atmosphere at 37 °C for 24 h to form a monolayer. A scratch was created by scraping a straight line using a 10 µL pipet tip. Cells were once washed with 1 mL of culture medium to remove debris and images were acquired at 0 and 24 h using a Nikon Diaphot 200 microscope, objective 10× (Nikon, Tokyo, Japan). To obtain the same field during image acquisition, the markings were created close to the scratch using an ultrafine marker.

The wound area was measured using ImageJ software following the protocol described by [35]. The rate of cell migration was calculated based on the change in % area covered with cells between time 0 and 24 h.

### 2.12. Statistic Analysis

Statistical analysis was performed in Prism 8.4 (GraphPad, San Diego, CA, USA). For the comparison of the control and prostate cancer cell line, *t*-test was applied. For multiple comparisons of cell lines overexpressing control vector or vector with mGPDH, one-way ANOVA and Bonferroni’s correction were used.

## 3. Results

### 3.1. Energetics of Prostate Cancer Cell Lines

To compare metabolic phenotype of the control epithelial cell line PNT1A and prostate cancer cell line LNCaP, we analyzed intact respiration and lactate production with glucose as a substrate using the Seahorse XFe analyzer. As can be seen from Figure 1A, oxygen consumption rate (OCR) was significantly increased in LNCaP cells compared to PNT1A (by 32%), while extracellular acidification rate (ECAR) was not different between the two cell lines. A different metabolic setting is best apparent from calculation of the OCR/ECAR ratio, which can reflect changes in both mitochondrial oxidative metabolism and glycolysis. The OCR/ECAR ratio was increased more than two-fold in LNCaP cells (Figure 1B). Since we observed higher mitochondrial respiration in LNCaP, we also checked for the content of individual complexes of oxidative phosphorylation (OXPHOS) apparatus by means of Western blot quantification. Here the content of representative subunits of OXPHOS complexes I–V (Appendix A) did not differ between PNT1A and LNCaP cells. Hence, we focused on mGPDH, since it represents alternative entry point of electrons into the respiratory chain. Indeed, we found that the enzyme activity was two-fold higher in the carcinoma cell line (Figure 1C). To quantify the mitochondrial reactive oxygen species (ROS) production in intact cells, we used fluorescent probe CM-H_2_DCFDA. Compared to the control cell line, the basal ROS production was profoundly increased in the prostate cancer cell line. A significant part of the ROS production was prevented by uncoupler (FCCP) indicating that a substantial part originates from the mitochondrial respiratory chain and is dependent on the mitochondrial membrane potential (Figure 1D). In accordance with these results, a pronounced difference was observed in the mGPDH content (Figure 1E) that was increased in LNCaP cells by one order of magnitude compared to the control cell line.

### 3.2. mGPDH Processing

Interestingly, we noticed that the mGPDH antibody detected two forms of the protein, further denoted as GP_high_ for the higher molecular weight band and GP_low_ for the lower molecular weight (MW) form. Strikingly, the proportion between GP_high_ and GP_low_ significantly differed between cell lines—GP_low_ content was higher in the prostate cancer cell line, representing 19.4% ± 9.3% in PNT1A and 58.0% ± 19% in LNCaP of the total mGPDH content (Figure 1F). We hypothesized that changes in mGPDH biogenesis and capacity could play an important role in the metabolic adaptation of prostate carcinoma.

Aiming to decipher the identity of the two mGPDH forms, we first estimated the size difference between GP_high_ and GP_low_ using SDS-PAGE/Western blotting and the Molecular Weight Analysis tool (from the Image Lab package). The analysis of five different LNCaP samples revealed that the difference between the two bands was 4.55 ± 0.16 kDa. We hypothesized that the two forms can either represent a differential post-translational modification (PTM) or proteolytic processing of the enzyme.

To search for possible PTMs in an unbiased way, we transfected HEK293 cells with mGPDH-FLAG (C-term) to subsequently enrich mGPDH via immunoprecipitation on anti-FLAG beads. Possible modifications of tryptic peptides were analyzed by the dependent peptides algorithm in MaxQuant, searching for modified peptides that are derived from an already identified “base” peptide. All possible hits are summarized in Appendix A. Apart from PTMs associated with protein damage (deamidation, oxidation), no other prominent modifications were identified and none of those identified ones could explain the observed MW shift. However, we observed prevalent modification of the tryptic peptide AADCISEPVNR starting at position 42. Here a peptide truncated by one alanine represented 38% of the total, suggesting the protein processing after position 42.

Proteolytic processing of mitochondrial proteins is mainly carried out as cleavage of the targeting presequence. Since no crystal structure is available for mGPDH, topology depends on prediction algorithms. While UniProt [36] predicts 42 amino acid long mitochondrial targeting peptide, other targeting peptide predictors such as Phobius [37] indicate a presequence consisting of 27 amino acids (Figure 2A). This would represent a mass of 4.509 kDa for 42 amino acid presequence and 2.678 kDa for 27 amino acid one (calculated by the Compute pI/MW tool [38]) (Figure 2B). Presumably, the observed MW difference between GP_high_ and GP_low_ would represent a precursor and mature form of mGPDH. To decipher the length of cleaved peptide experimentally, we overexpressed the *GPD2* gene FLAG-tagged at different positions in HEK293 cells. The FLAG tag was localized either on the C terminus (C-term), N terminus (N-term), after Phe27 (27AA) or Ala42 (42AA) amino acid as described in the methods. The SDS-PAGE/Western blot analysis of FLAG-tagged forms of mGPDH demonstrated that the FLAG antibody recognized the 42AA processed form but not N-term and 27AA variants, indicating that the cleavage site is between Phe27 and Ala42 (Figure 2B). Again, this provides evidence in favor of the longer predicted presequence.

### 3.3. Role of IMMP2L

In mice with reproductive failure (generated by a random transgenic insertional mutagenesis), mitochondrial membrane peptidase 2-like (IMMP2L) was identified as a peptidase that processes the mGPDH protein [39]. Recently, also human IMMP2L peptidase was associated with mGPDH processing during senescence [40]. Therefore, we tested the level of the peptidase in prostate benign and malignant cell lines. As shown in Figure 2D, the level of IMMP2L protein was significantly increased in the prostate cancer cell line. Thus, increased IMMP2L levels could be responsible for more profound processing of mGPDH in the LNCaP cells.

Since IMMP2L is localized in the mitochondrial intermembrane space, it can only cleave mGPDH in this compartment. Therefore, in order to determine the compartment where mGPDH processing occurs, we fractionated LNCaP cells collecting whole cell homogenates, soluble cytosolic fraction, and isolated nuclei and mitochondria (Figure 3A). Both GP_high_ and GP_low_ were present in identical proportions in isolated mitochondria as well as in whole cell homogenates. No traces of mGPDH protein were detected in the cytosolic fraction. The weak mGPDH signal in the isolated nuclei is likely due to mitochondrial contamination of this fraction as citrate synthase is also detected. This result indicates that the unprocessed form of the enzyme is transported to mitochondria where the cleavage of the mGPDH targeting presequence occurs. This again supports the role of IMMP2L in this process, since its yeast homologue Imp2p cleaves the intermembrane space-sorting signal [39].

### 3.4. Kinetics of mGPDH Processing

As a next step, we analyzed kinetics of mGPDH processing. Therefore, we performed the protein turnover analysis during 24 h after emetine-induced inhibition of proteosynthesis in LNCaP prostate cancer cells (Figure 3B). In the steady state (time 0 h), we observed equal proportion of GP_high_ and GP_low_ forms. However, during the 24 h period, the unprocessed form (GP_high_) gradually disappeared while the content of the mature protein (GP_low_) increased, at the end representing more than 80% of the total mGPDH signal. We further evaluated the kinetics of mGPDH maturation in HEK293 cells transiently transfected with C-terminally FLAG-tagged GPD2 (C-term), where FLAG tag remains also in the processed form (see Figure 2C) and can be visualized by SDS-PAGE Western blotting. This experiment revealed that the maturation of recombinant mGPDH is rather slow in control cells, since only the nascent form was present 16 h after transfection, while the processed form was appearing gradually after 48 and 72 h (Figure 3C). As can be seen from Appendix A, IMMP2L levels do not respond to GPDH-FLAG transfection and remain the same for all assessed timepoints. It is therefore likely, that insufficient IMMP2L content in HEK293 cells limits the speed of recombinant mGPDH-FLAG processing. Additionally, continuous synthesis of new nascent GPDH-FLAG during the whole time-course contributes to a much milder decrease in the content of GP_high_ (red line), compared to the emetine experiment, where blocking of proteosynthesis totally abolishes the flux of a new nascent protein.

### 3.5. Native Forms of mGPDH

Unlike the mitochondrial matrix or inner membrane proteins, where cleavage of the targeting presequence represents an essential part of their import, the intermembrane space (IMS) proteins are not typically processed in this way [41]. Therefore, we aimed to explore the physiological role of mGPDH processing. To elucidate whether N-terminal peptide cleavage has consequences for mGPDH enzyme activity, we analyzed native forms of mGPDH present in the membrane by means of high-resolution clear native electrophoresis (hrCNE) combined with detection of both mGPDH content and in-gel activity. In HEK293 cells overexpressing an untagged form of mGPDH (HEK oe), in-gel activity in hrCNE separated proteins revealed several active mGPDH forms (Figure 4A). The size of the lowest-migrating form was assessed by parallel detection of the succinate dehydrogenase complex of a known size (approximately 140 kDa), whose migration pattern was identical to the mGPDH band. Since the mGPDH monomer had 70 kDa, the detected active form most likely represents the mGPDH dimer. Importantly, this dimeric band was also prominent for endogenous mGPDH in LNCaP prostate cancer cells (Appendix A). The higher migrating mGPDH bands represented more complex enzyme oligomers.

To decipher the composition of mGPDH native forms, we performed the two-dimensional analysis (hrCNE/SDS-PAGE) in both cell lines. Interestingly, the migration pattern of nascent (GP_high_) and processed (GP_low_) mGPDH differs and the active entities contained mainly the processed form in HEK oe (Figure 4C), as well as in LNCaP cells (Appendix A). This was most prominent for the enzyme dimer, where solely the processed (GP_low_) form was detected. These data indicate that presequence cleavage is a prerequisite for subsequent assembly into the dimer, which represents the active form of mGPDH.

### 3.6. mGPDH Content Modulates Cell Migration

In order to uncover the role of mGPDH in prostate cancer cells, we overexpressed mGPDH or empty vector in LNCaP cells. We observed an increased protein level of the mGPDH protein level (Figure 5A) as well as mGPDH enzyme activity (Figure 5B). Neither mGPDH content nor its activity were affected by the transfection, as can be seen from identical values for either parental cells or cells transfected with an empty vector (Figure 5A,B).

Further, we tested if increased mGPDH expression may have an impact on the cell migration ability (wound healing) of prostate cancer cells. We performed an in vitro scratch assay in LNCaP cells stably transfected with vector expressing mGPDH or a control vector and found that increased mGPDH content led to an increased rate of the cell migration in LNCaP cells, which was significantly increased by 25%, compared to its empty vector counterpart (Figure 5C).

## 4. Discussion

During tumorigenesis, cells undergo distinct adaptations of cellular metabolism in order to survive, proliferate and disseminate [8]. Unlike many cancer types, prostate cancer does not increase glucose uptake (i.e., FDG-PET fails to detect primary tumors [42]). This necessitates the involvement of other substrates including lactate degradation or branched-chain fatty acid oxidation. However, it is still questionable, which is the major substrate utilized in vivo in prostate cancer cells [43]. Our results revealed that energy metabolism, namely the oxygen consumption rate, of LNCaP prostate cancer cells is increased compared to the prostate epithelial control cell line PNT1A (Figure 1A,B). This is in accordance with findings of Dueregger et al. [13] who observed higher basal oxidative respiration (baseline respiration or ROUTINE state) in intact prostate cancer cells (LNCaP, PC3) compared to their healthy counterparts (RWPE1) supplied with glucose using Oroboros Oxygraph. The same study reported, that prostate cancer cells had lower capacity for the oxidation of fatty acids and complex I-dependent substrates and this was substituted by increased respiration fueled with complex II-dependent substrate—succinate [13]. Interestingly, another study revealed that the activity of mitochondrial glycerol-3-phosphate dehydrogenase (mGPDH), an alternative dehydrogenase that can feed the respiratory chain with electrons, is increased in prostate cancer cells [17]. We have also determined a profound increase in ROS formation in intact LNCaP cells when compared to PNT1A. This is in line with a previous observation, when respective cells were supplied with exogenous succinate or glycerophosphate [17]. At least in part, this increase can be attributed to mGPDH, since it has been demonstrated to be an important direct source of electron leak [19,20] as well as indirectly contribute to electron leak on other complexes of respiratory chain [44]. Increased ROS production is in general associated with numerous cancer cell models and these ROS can also act as a proliferative signal. This can potentially represent yet another functional role of mGPDH in malignant cells in addition to reoxidation of cytosolic NADH through glycerophosphate shuttle [19]. Therefore, we analyzed the protein content of OXPHOS complexes (complex I-V) as well as mGPDH by Western blotting and found, that unlike canonical OXPHOS enzymes (Appendix A), mGPDH level increased significantly in cancer compared to benign cells (Figure 1E). We also observed higher specific activity of mGPDH supporting the importance of the enzyme in prostate cancer cells (Figure 1C).

Mitochondrial glycerol-3-phosphate dehydrogenase is a part of the glycerophosphate shuttle and its primary role is to reoxidize cytosolic NADH produced by glycolysis [19]. However, mGPDH expression is highly variable across the tissues [45] and differs among cancer types [17,46]. Its substrate glycerol-3-phosphate (G3P) may be derived from triglyceride catabolism. Conversely, this intermediate can be also used for phospholipid and triglyceride synthesis [19,22], as it was shown that G3P oxidation is suppressed by fatty acids and acyl-CoA esters [47]. Since mGPDH activity is rate limiting for the flux through the glycerophosphate shuttle, it is the mGPDH content that has to be increased in order to enhance cytosolic NADH reoxidation [19]. In this regard, the G3P pool size does not have to be changed, as it cycles within the glycerophosphate shuttle. Interestingly, we observed two distinct forms of mGPDH (GP_high_ and GP_low_) with a mass difference of approximately 4.5 kDa (Figure 2B) and their proportion significantly differed between benign and malignant cell lines (Figure 1F). Considering the important role of mGPDH in cellular metabolism, we hypothesized that this regulation may represent a possible mechanism of metabolic adaptation during tumorigenesis.

The shift in protein size by 4.5 kDa may be caused by either specific post-translational modifications (e.g. glycosylation) or by proteolytic processing of the protein. Proteomic analysis of mGPDH in HEK293 cells revealed no relevant post-translational modifications (PTMs), which is also in line with published HTS studies for various PTMs, where only a relevant hit reported for mGPDH is phosphorylation on Tyr601 in the brain [48,49]. Proteolytic processing of mGPDH associated with its import into intermembrane space (IMS) is therefore a much more relevant reason for the observed MW shift. In general, protein targeting into IMS can be achieved via two processes (reviewed in [41,50]). Proteins lacking specific presequence typically follow redox-dependent mitochondrial IMS assembly pathway (Mia/Erv40) and contain twin CX9C or CX3C motifs to form intramolecular disulfide bridges. The second class of proteins contain a bipartite N-terminal sequence—the first part represents the mitochondrial targeting presequence and the second hydrophobic stop-transfer signal, which leads to lateral insertion into the inner membrane by the TIM23 complex. Consequently, cleavage by IMS proteases releases the mature protein into IMS. Since there is only one weak CX14C motif in mGPDH, it is much more likely that it follows the stop transfer pathway. Based on homology with bacterial GlpD [51], mGPDH is the monotopic membrane protein and in this regard unique in IMS (other mitochondrial monotopic proteins are matrix oriented and their insertion occurs from the matrix side [52]). Hydrophobic patch anchoring mGPDH in the membrane is not inserted via TIM23 and cannot in itself act in the stop transfer mechanism, this must be the role of the (cleaved) presequence. Indeed, within the 42AA presequence, which we identified in this work, both TMHMM [53,54] and TMpred [55] algorithms predict the transmembrane helix (positions 7-29 or 9-28 respectively), which can be laterally inserted into the membrane. As expected, the presequence part of mGPDH does not show any homology with bacterial GlpD, another clue that it evolved solely as an insertion mechanism for the mitochondrial enzyme.

If mGPDH undergoes presequence cleavage by IMS protease, the second issue to resolve remains, which protease is responsible for this process. In this regard, mitochondrial IMMP2L peptidase has been associated with processing of this protein in mice [39] and humans [40], but also other substrates of IMMP2L have been indicated, namely cytochrome c1 [39] and the apoptosis-inducing factor (AIF) [40]. In agreement with these publications, our experiments showed that mGPDH had only one cleavage site and that it was after Ala42 (Figure 2C) most probably by IMMP2L peptidase that we also found significantly increased in the prostate cancer LNCaP cell line (Figure 2D). The localization of mGPDH cleavage in mitochondria was also supported by the cancer cells fractionation experiment since the proportion of mGPDH forms were identical in mitochondria and whole cell homogenates and no mGPDH protein was observed in the cytosol (Figure 3A).

Impairment of IMMP2L in mouse led to decreased processing of cytochrome c1 (part of complex III of respiratory chain) and subsequently to a slight but significant decrease in enzyme activity of complex I+III (NADH:cytochrome c oxidoreductase) indicating the functional relevance of cytochrome c1 processing [39]. Therefore, we hypothesized that also in the case of mGPDH cleavage of the N-terminal part is essential for its function. Indeed, upon protein synthesis inhibition by emetine (0–24 h), we observed a continuous increase of the proportion of mature to nascent form (Figure 3B). Maturation itself seems to be a rather slow process, because transient transfection of HEK293 cells by vector expressing mGPDH tagged with FLAG at C-terminus showed sequential protein maturation—16 h after transfection the nascent form was detected, the mature form was only present after 48 h and its relative amount increased after 72 h (Figure 3C). Indeed, slow induction of mGPDH activity had been described previously. Thus, upon triiodothyronine induction of mGPDH in liver, maximum activity was observed at the 24 h timepoint, long after the peak in the mRNA level [28].

It is also unclear, whether both preprotein (GP_high_) and mature mGPDH (GP_low_) are enzymatically active. Measurements of isolated GP:DCPIP activity with DCPIP as an artificial acceptor in control mice versus mice with IMMP2L mutation, and thus defective mGPDH processing, implicated, that both forms are active [56]. To the contrary, data from MIR-90 cells with IMMP2L overexpression associated higher mGPDH processing with higher glycerol-3-phosphate levels, implicating the role of processing on enzyme activity [40]. In this study, we clearly demonstrated that the enzyme dimer consisted solely of the processed form (GP_low_; Figure 4 and Appendix A). As suggested from the crystallographic protein structure analysis from mGPDH bacterial homologue GlpD, the dimer is a physiologically relevant enzyme structure [19,51] implying functional significance of mGPDH maturation. It is therefore possible, that even the nascent GP_high_ form is capable of electron transfer to DCPIP [56], but in the context of the respiratory chain, only the mature processed GP_low_ represents the active form of the enzyme. Indeed, even for solubilized mGPDH, significant GP:DCPIP activity was reported [57], indicating that this activity can be observed also for the enzyme outside of the respiratory chain context.

It has been shown that cellular metabolism is involved in cell migration and eventually in metastasis, and that addiction of cancer cells on mitochondria depends on the cancer type (summarized by [43,58,59]). Being at the crossroad of glycolysis, lipid and oxidative metabolism [19,60], mGPDH controls important metabolic functions and may be involved in metabolic adaptations during tumorigenesis. Interestingly, cell migration measured by an in vitro scratch assay in malignant cancer cells was increased in cancer cells overexpressing mGPDH compared to cancer cells transfected with the control vector (Figure 5).

In summary, improved cell migration in cancer cells overexpressing mGPDH indicates the functional importance of the enzyme in cancers with high expression of the enzyme. We suggest that the activity may be regulated not only at the level of the mGPDH content, but also by its processing since the functional dimer is composed solely of the mGPDH mature form. Furthermore, the processing is most probably mediated by inner membrane protease IMMP2L by the cleavage of N-terminal (42 amino acids). Unlike IMMP2L, mGPDH is highly tissue specific and its impairment would not affect all tissues, which means that mitochondrial glycerol-3-phosphate dehydrogenase represents a suitable target for cancer therapy.

## 5. Conclusions

In this paper, we thoroughly analyzed posttranslational processing of mGPDH and showed that the 42 amino acid presequence is cleaved from N-terminus during mGPDH biogenesis. Only the processed form is part of the mGPDH dimer, which represents the prominent functional enzyme entity. We have also demonstrated, that mGPDH overexpression enhances the wound healing ability in prostate cancer cells. As mGPDH is at the crossroad of glycolysis, lipogenesis and oxidative metabolism, regulation of its activity by intramitochondrial processing might serve in cellular metabolic adaptations.

## Figures and Tables

**Figure 1 cells-09-01764-f001:**
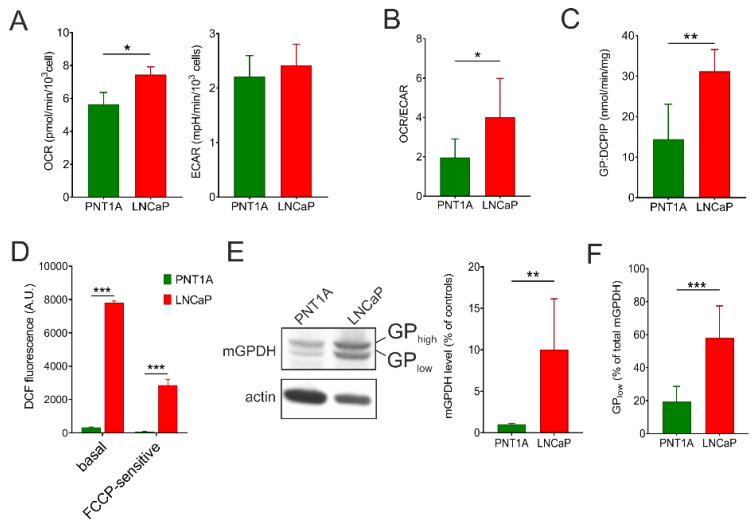
Energy metabolism and mGPDH content in prostate cancer cells. The oxygen consumption rate (OCR) and extracellular acidification rate (ECAR) (**A**), as well as OCR/ECAR ratio (**B**) in prostate cancer cells (LNCaP) compared to control epithelial cell line (PNT1A). Values were calculated from the rates in basal conditions, i.e. at the presence of 10 mM glucose, and determined by a Seahorse XFe analyzer (*n* = 5). (**C**) Enzyme activity of mGPDH measured spectrophotometrically using 10 mM glycerol-3-phosphate as a substrate (*n* = 6). (**D**) ROS generation in intact LNCaP cells compared to control PNT1A measured by the CM-H_2_DCFDA probe. To determine the FCCP-sensitive portion of ROS production, 1 μM uncoupler was used. (**E**) Cell lysates (15 μg protein) were separated on SDS-PAGE and mGPDH content was analyzed by Western blotting using a specific antibody against mGPDH, actin was used as a loading control. Representative blot of 5 independent experiments is depicted. Antibody signals were quantified densitometrically as the total mGPDH levels normalized to actin levels and the results are expressed as % of control values. (**F**) Processing of mGPDH was determined densitometrically as a ratio of the lower band and total mGPDH content (*n* = 5). Data represent the means ± S.D., * *p* < 0.05, ** *p* < 0.01, *** *p* < 0.001.

**Figure 2 cells-09-01764-f002:**
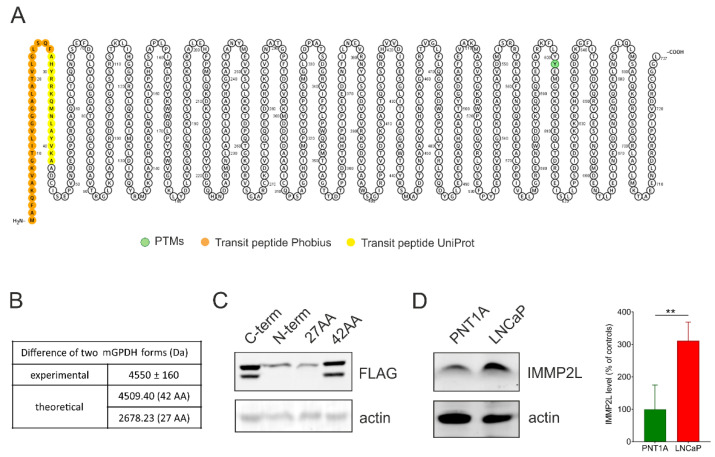
Analysis of mGPDH forms. (**A**) mGPDH topology by the Protter visualization tool. Predicted phosphorylation (green) and targeting sequences by Uniprot (yellow) and Phobius (orange) are depicted. (**B**) Table of experimental and calculated differences of two mGPDH forms. Experimental difference (*n* = 5) was determined by the Molecular Weight Analysis tool (Image Lab software, Bio-Rad). (**C**) Cell lysates (50 μg protein) of HEK293 cells overexpressed with FLAG in different places of the GPD2 sequence were separated on SDS-PAGE (Hoefer System) and mGPDH forms were analyzed by Western blot using a specific antibody against FLAG (*n* = 3). C-term—GPD2 with FLAG tag at the C-terminal, N-term—GPD2 with FLAG following initial start codon (N-term), 27AA—GPD2 with FLAG tag following 27th amino acid and 42AA—GPD2 with FLAG tag following 42nd amino acid. (**D**) Cell lysates (15 μg protein) of control PNT1A and cancer (LNCaP) cells were separated on SDS-PAGE and IMMP2L peptidase content was analyzed by Western blotting using a specific antibody (*n* = 5). Actin was used as a loading control (**C–D**). Data represent the means ± S.D., ***p* < 0.01.

**Figure 3 cells-09-01764-f003:**
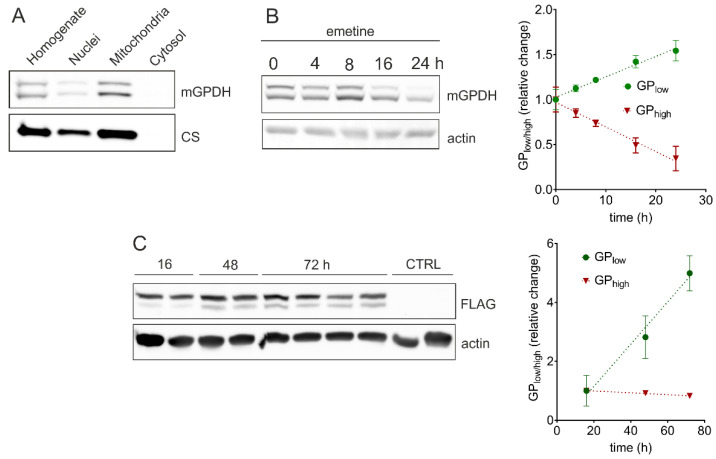
Western blot analyses of mGPDH processing and biogenesis. All samples were separated on SDS-PAGE and analyzed by Western blotting. (**A**) Lysates (30 μg protein) of different fractions of LNCaP cells were analyzed with specific antibodies against mGPDH and citrate synthase. SDS-PAGE and Western blot analysis of (**B**) cell lysates (30 μg protein) of LNCaP cells after inhibition of proteosynthesis by emetine (100 μg/mL) using a specific antibody against mGPDH and (**C**) cell lysates (20 μg protein) of HEK293 cells transiently transfected with a FLAG tag on C-term using a specific antibody against FLAG. Actin was used as a loading control. Quantified data represents a relative change of the GP_low/high_ form of three independent experiments and is expressed as means ± S.D.

**Figure 4 cells-09-01764-f004:**
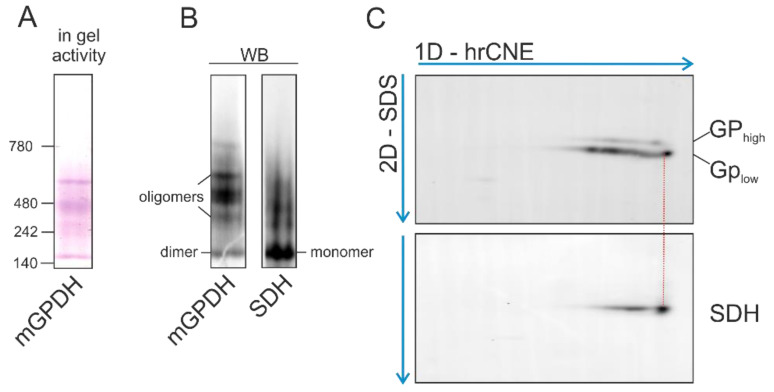
Analysis of mGPDH native forms. HEK293 overexpressing mGPDH (HEK oe, untagged form) were solubilized by digitonin (2 g/g) and samples (40 μg protein) were separated on high-resolution clear native electrophoresis (hrCNE, 5–13%) and either stained in the gel for mGPDH activity (**A**) used for Western blot detection of mGPDH or SDHA subunits using specific antibodies (**B**) or strips were subjected on SDS-PAGE (12%) and analyzed by Western blot as on hrCNE (**C**).

**Figure 5 cells-09-01764-f005:**
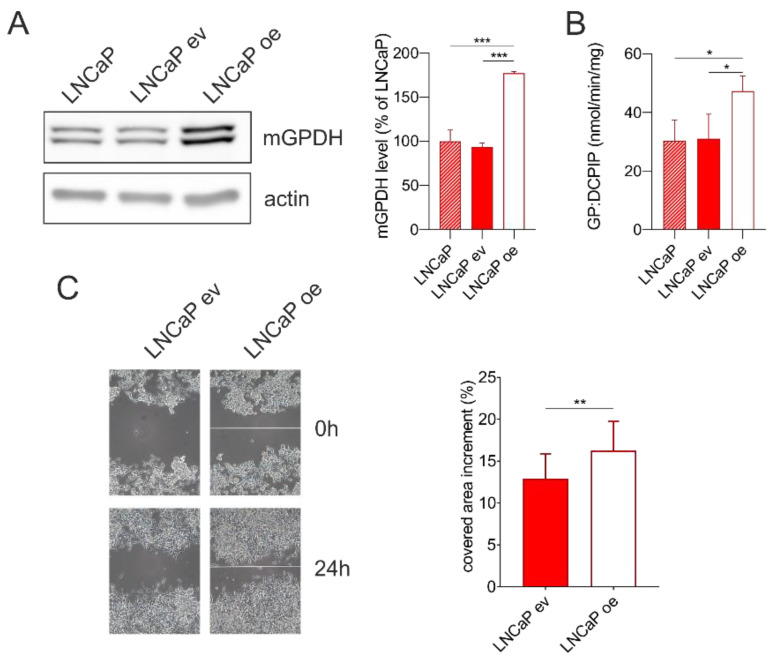
Wound healing assay in LNCaP cells without transfection, stably transfected with a control pcDNA3.1(+) vector (LNCaP ev) or vector containing mGPDH (untagged form; LNCaP oe). (**A**) Cell lysates (15 μg) were separated on SDS-PAGE and analyzed by Western blotting using antibodies against mGPDH and actin. A representative Western blot and quantification of total mGPDH amount is shown (*n* = 3). (**B**) Enzyme activity of mGPDH measured spectrophotometrically in cell lysates using glycerol-3-phosphate as a substrate (*n* = 3). (**C**) Analysis of cell migration by in vitro scratch assay. Images were acquired at 0 and 24 h. Representative images and quantification of covered area increment (*n* = 15). Data represent the means ± S.D., * *p* < 0.05, ** *p* < 0.01, ****p* < 0.001.

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
