# Peer review of "Role of Mitochondrial Glycerol-3-Phosphate Dehydrogenase in Metabolic Adaptations of Prostate Cancer"

_cells, 2020, doi:10.3390/cells9081764_

Round 1
Reviewer 1 Report
The manuscript of Pecinova et al. is focussed on the mitochondrial inducible enzyme GPDH, showing novel details of its processing. The mGPDH activity is higher in prostate cancer cell line LNCaP in comparison with the prostate epithelial control cell line PNT1A, due to enhanced protein level. In particular, two distinct forms of mGPDH with slightly different mass (high and low) were detected, the low form being more abundant in cancer cells than in controls. Only the processed form (low) seems to be part of the active mGPDH dimer, supporting the notion that prostate carcinoma, in contrast with other cancer types, undergo metabolic adaptations involving mitochondria as a main energy source for sustaining rapid cell growth.
The experimental strategy is well-designed and several rather sophisticated biochemical approaches have been utilized (clear native electrophoresis and western blot; mGPDH In-gel-activity, 2D CNE/SDS-PAGE and Western blot, time-course of in vivo mGPDH processing). Furthermore, the manuscript is clearly written.
However, I found some weaknesses in the data reported in last part of the study, where the authors tried to translate their results into the field of tumorigenesis. I point out that this last part deserves some attention.
Major point
My concern deals with the data shown in fig.5. The western blotting reported in Fig.5A shows no difference in the amount of mGPDH between LNCaP and PNT1A cells transfected with empty vector. This result is completely different from that shown in fig. 1A, where the difference between the two (not transfected) cell lines is quite strong. It seems that the presence of empty vector has some effect. How do the authors explain the difference between fig.1 and 5? This has to be commented.
Then the authors “hypothesized that higher expression of mGPDH may be beneficial for prostate cancer cells and enhance their cell migration ability”. Therefore, the proliferation efficiency of the cell lines was measured (Fig. 5B). The intensity of mGPDH overexpression in the two cell lines was relatively modest (less than 20% increase in protein content) explaining the very weak effect of mGPDH overexpression on cell proliferation. The results of the scratch assay in LNCaP and PNT1A (not transfected) cells has to be shown as a positive control, given the highly increased endogenous mGPDH level (10-fold) and activity of the LNCaP cells.
Minor point
It has been described also by the same authors that the activity of mGPDH is associated with significant overproduction of ROS (Mráček et al, BBA 2013; Chowdhoury BBRC 2006). A comment on this should be added, also considering that ROS are also signals promoting proliferation.
Reviewer 2 Report
The paper of Pecinová et al. is an interesting contribution to the growing field of intermediary metabolism of tumors. Authors overexpressed mitochondrial glycerophosphate dehydrogenase (mGPDH) in normal human prostate epithelial cell line PNT1A and in prostate cancer cell line LNCaP. It was found that overexpression of mGPDH is associated with increased migration ability in LNCaP cells. Authors extensively characterized the forms and maturation of mGPDH in HEK cells and in cells of prostatic origin. They also described that the level of mitochondrial membrane protease IMMP2L - playing a role in the processing of mGPDH - is higher in LNCaP cells than in normal epithelia.
Nevertheless the referee has some concerns about the interpretation of the results. The most important problem is:
There was no difference found in the wound healing assay between non malignant and malignant control cell lines in terms of the control area (in both controls it is around 12-13%). If this assay is a model of migration ability, therefore correlation is expected between migration ability and metastasis formation I would expect a significant difference.
The transfection (overexpression) of mGPDH is associated with increased tendency of migration in the cancer cell line. However, the total amount of mGPDH is only about 20% higher than in the control transfected cells (Fig. 5A). In Fig.1 C, D and E authors showed that mGPDH activity is more than two times, and immunreactivity is more than three times higher in LNCaP than in PNT1A cells.
It is difficult to see the causality between the mGPDH and the wound healing (cell migration) test’s results.
In the literature three metabolic patterns are associated with the prostatic cells. The normal ones produce citrate and rather glycolytic. The cancer cells have a highly oxidative metabolism and their glucose consumption is relatively low, but they utilize lactate released from the stromal cells. The metastatic cancer is highly glucose dependent. Authors I think should discuss the origin of glycerophosphate for the mGPDH enzyme.
Minor comments:
Fig.4. Authors mention D part of the figure. I can not see Fig. 4D.
Round 2
Reviewer 1 Report
The revised version of the manuscript is suitable for publication.
Reviewer 2 Report
Authors answers are satisfactory. The MS was significantly improved